# Whole Genome Sequencing of *Aggregatibacter actinomycetemcomitans* Cultured from Blood Stream Infections Reveals Three Major Phylogenetic Groups Including a Novel Lineage Expressing Serotype a Membrane O Polysaccharide

**DOI:** 10.3390/pathogens8040256

**Published:** 2019-11-22

**Authors:** Signe Nedergaard, Carl M. Kobel, Marie B. Nielsen, Rikke T. Møller, Anne B. Jensen, Niels Nørskov-Lauritsen

**Affiliations:** 1Department of Clinical Microbiology, Aarhus University Hospital, DK-8200 Aarhus N, Denmark; nedergaardsigne@gmail.com (S.N.); kobel@pm.me (C.M.K.); mabanielsen@health.sdu.dk (M.B.N.); torsbjerg-92@hotmail.com (R.T.M.); abj@dent.au.dk (A.B.J.); 2Bioinformatics Research Centre, Aarhus University, DK-8000 Aarhus C, Denmark; 3Department of Clinical Biochemistry and Pharmacology, Odense University hospital, DK-5000 Odense C, Denmark; 4Department of Dentistry, Aarhus University, DK-8000 Aarhus C, Denmark

**Keywords:** taxonomy, core and accessory genes, average nucleotide identity, principal component analysis

## Abstract

Twenty-nine strains of *Aggregatibacter actinomycetemcomitans* cultured from blood stream infections in Denmark were characterised. Serotyping was unremarkable, with almost equal proportions of the three major types plus a single serotype e strain. Whole genome sequencing positioned the serotype e strain outside the species boundary; moreover, one of the serotype a strains was unrelated to other strains of the major serotypes and to deposited sequences in the public databases. We identified five additional strains of this type in our collections. The particularity of the group was corroborated by phylogenetic analysis of concatenated core genes present in all strains of the species, and by uneven distribution of accessory genes only present in a subset of strains. Currently, the most accurate depiction of *A. actinomycetemcomitans* is a division into three lineages that differ in genomic content and competence for transformation. The clinical relevance of the different lineages is not known, and even strains excluded from the species sensu stricto can cause serious human infections. Serotyping is insufficient for characterisation, and serotypes a and e are not confined to specific lineages.

## 1. Introduction

*Aggregatibacter actinomycetemcomitans* is a fastidious Gram-negative bacterium that inhabits the mucosal surfaces of humans and certain primates [1,2]. The species has attracted attention due to its association with periodontitis [3]. Particularly, a single serotype b clonal lineage is associated with a silent but aggressive orphan disease of adolescents that results in periodontitis and tooth loss [4]. Rather than being the causative agent of aggressive periodontitis, *A. actinomycetemcomitans* may be necessary for the action of a consortium of bacterial partners by suppressing host defences [5].

*A. actinomycetemcomitans* is a member of the HACEK group of fastidious Gram-negative bacteria (*Haemophilus*, *Aggregatibacter*, *Cardiobacterium*, *Eikenella* and *Kingella*), a recognized but unusual cause of infective endocarditis responsible for 1.4% to 3% of cases [2]. A recent population-based study of the incidence of HACEK bacteraemia in Denmark identified 147 cases corresponding to an annual incidence of 0.44 per 100,000 population [6]. A retrospective study from New Zealand with 87 cases of HACEK bacteraemia confirmed a strong association with infective endocarditis, although the association with endocarditis ranged from 0 of 11 cases (*Eikenella corrodens*) to 18 of 18 cases (*A. actinomycetemcomitans*) [7]. 

Traditional classification of *A. actinomycetemcomitans* into serotypes is based on the chemical structures of the outer membrane O polysaccharide. Other studies have addressed the species’ population structure by subjecting selected strains to multilocus enzyme electrophoresis [8], 16S rRNA gene sequencing [9], or multilocus sequence typing [10]. All methods identified an outgroup consisting of a subset of serotype e strains, but the grouping of serotypes was not consistent between methods. Finally, whole genome sequences (WGSs) of *A. actinomycetemcomitans* have become available. The first comparison of 14 strains found two major groups composed of serotypes a, d, e, plus f, and b plus c, respectively, while a serotype e strain outgroup showed a conspicuous lack of the cytolethal distending toxin gene cluster [11]. Jorth and Whiteley added three additional WGSs and calculated average nucleotide identity (ANI); strains within the two major groups were ~99% identical, while comparisons of strains belonging to separate groups disclosed significant differences (ANI of ~97%), and the outgroup strain was positioned outside the recommended species boundary [12]. The most recent comparison of WGSs included non-*actinomycetemcomitans Aggregatibacter* strains and was restricted to 397 concatenated core genes. The analysis suggested a division of the species into five clades: clade b (serotype b), clade c (serotype c), clade e/f (serotypes e and f), clade a/d (serotypes a and d), and clade e′ (outgroup serotype e strains) [13].

The few studies that have characterised *A. actinomycetemcomitans* from cases of bacteraemia were limited to serotyping [14], or serotyping supplemented with arbitrarily primed PCR [15]. Here, we present WGSs of 29 Danish bacteraemia isolates. We include five additional oral strains to characterise a novel group within the species designated lineage III.

## 2. Results

Twenty-nine blood stream isolates of *A. actinomycetemcomitans* were identified as part of an investigation of Danish HACEK bacteraemia cases [6]. Serotyping by PCR identified seven serotype a, 11 serotype b, 10 serotype c, and one serotype e strain—this distribution is similar to the observed prevalence among oral strains in Scandinavia ([16] and references therein). Comparison of WGSs did, however, show that the single serotype e strain (PN_561) was unrelated to common isolates of this serotype, but clustered with the clade e’ outgroup [13]. Moreover, one serotype a strain (PN_696) was unrelated to the other six isolates of this serotype and did not cluster with any *A. actinomycetemcomitans* WGSs present in the public databases. We tried to identify further isolates of this peculiar genotype. An early characterisation by multilocus enzyme electrophoresis of 97 strains isolated over a period of 45 years identified four minor groups that deviated from the two major divisions [8]. By WGSs, the three strains of division V (HK_907, HK_973, and HK_974) clustered with the aberrant serotype a blood stream isolate. An investigation of stored serotype a strains revealed two additional members of this lineage (K51, HK_1710), and these five oral isolates are included in the comparison.

The neighbour-joining comparison of core gene sequences of 35 study strains (including the type strain NCTC 9710 of serotype c) plus seven selected reference sequences downloaded from the public databases is shown in Figure 1. One blood stream isolate (PN_561) plus a reference sequence comprise the clade e’ outgroup that is used to root the tree. Although serotype b and serotype c strains are distributed in two separate branches, the overall population structure of the species consists of three separate lineages.

### 2.1. Delineation of the Species

Serotype e strain PN_561 was cultured from a case of bacteraemia in 2005. The patient underwent aortic valve replacement for infective endocarditis, and the bacterium was also cultured from the removed valve. The strain was identified as *Actinobacillus (Aggregatibacter) actinomycetemcomitans* based on selected phenotypic tests; re-examination using matrix-assisted laser desorption/ionization time-of-flight (MALDI-TOF) mass spectrometry confirmed the identification with a log-score value above 2. The clinical case demonstrates the aggregative potential of the clade e’ outgroup, and the WGSs reveal the presence of a *tad* cluster and leukotoxin operon, but a lack of the cytolethal distending toxin gene cluster (GenBank accession number VSEC00000000). We performed in silico DNA–DNA hybridisation (DDH) by use of the Genome-to-Genome Distance Calculator 2.1, which estimates the DDH values that would have resulted from classic hybridisation experiments [17]. The in silico DDH value for strain PN_561 versus the type strain NCTC 9710 was 54.7%, which is below the phylogenetic species boundary of 70% suggested by classic hybridisation [18]. Similar in silico DDH values were obtained for PN_561 versus selected reference strains from other serotypes (range: 54.2% (D18P-1, serotype f) to 55.4% (HK_1651, serotype b)). 

Average nucleotide identity (ANI) is a powerful method to estimate overall genome relatedness and is widely used as a substitute of the classic DDH methods. ANI is calculated for two genome sequences by breaking the genome sequence of the query strain into 1020-bp-long fragments. Then, nucleotide identity values for individual fragments of the query strain and the genome of the subject strain are calculated using the NCBI BLASTn program. Using OrthoANI [19], where both genomes are fragmented and only reciprocal BLASTn hits are included, strain PN_561 was 93.89% identical with the type strain of *A. actinomycetemcomitans*; restricting ANI calculation to 1146 concatenated core gene sequences gave an ANI value of 94.72%. An ANI threshold value of 95% is considered the species boundary [20,21], and the two clade e’ outgroup strains are excluded from further analysis.

### 2.2. Natural Competence and Genomic Characteristics of Lineages 

Twenty-eight invasive strains plus five oral strains of *A. actinomycetemcomitans* were tested for natural competence by plating on kanamycin-containing agar in the presence of donor DNA. All strains belonged to the three dominant serotypes a–c. In accordance with previous findings [22], only a subset of serotype a strains were competent for transformation, while serotype b and c were invariably noncompetent. Specifically, competence was associated with lineage II (strains PN_437, PN_559, PN_563, PN_567, and PN_688), while all strains of lineage III were noncompetent. Competence is a primary mechanism of horizontal gene transfer and DNA acquisition in bacteria, and it has previously been shown that competent strains of *A. actinomycetemcomitans* are, on average, 200,000 bp larger than noncompetent strains [12]. In accordance with this, the mean genome size of strains of lineage II was 2.26 Mb, while the mean genome size of invariably noncompetent strains of lineage I was 2.07 Mb. With an average genome size of 2.22 Mb, lineage III was closely related to the mean size of competent lineage II, but the range (2113–2316 Kb) did overlap the size of the largest genome in lineage I (strain PN_738; 2134 Kb). 

Analysis of competence genes in the six strains of lineage III revealed major disruptions within the essential repository (Figure 2). All strains carried large (11–28 Kb) mobile elements impeding *comM*, but no genome contained all 16 genes, and 31 of 84 identified competence genes were inactivated. It is possible that future analysis will reveal some competent strains of lineage III, but the investigated genomes indicate an ancestral noncompetent lineage, which questions the relationship between competence and genome size.

Excluding the two clade e’ outgroup strains from Figure 1 increased the number of core gene sequences from 1146 to 1357 but did not distort the dendrogram (not shown). The total number of genes identified by Prokka was 4631, and the distribution of accessory genes is of interest. We used principal component analysis (PCA) of the dichotomous presence/absence of orthologs, analysing 3274 genes present in 1–39 of 40 strains. Figure 3A shows a scatterplot of this accessory genome of 40 *A. actinomycetemcomitans* strains, represented by the two principal components that account for a major part of the variance in the gene presence/absence matrix (Figure 3B). Analysis of the accessory genome supports the division of the species into three distinct lineages. The first principal component (PC) primarily serves to separate lineage I from lineages II and III, while PC2 dissociates all three lineages (Figure 3A). Indeed, nine of the 10 annotated genes (excluding hypothetical genes) with the highest loadings in PC1 were only present in 23 strains of lineage I, while nine of the 10 annotated genes with the highest negative loadings were predominantly associated with 17 strains of lineage II plus III (range 15–18). For PC2, the highest loadings were associated either with 34 strains of lineage I plus II, or with six strains of lineage III, while the highest negative loadings were more unevenly distributed among lineages (not shown). 

By scatterplot of PC1 vs. PC2, strains of lineage III appear more diverse than those of lineage II, which are more diverse than those of lineage I (Figure 3A); this may, in part, be caused by the decreasing number of strains included in the lineages. The accessory gene content of serotype g strain NUM4039 was the most divergent in lineage II, followed by serotype a strain D7S-1 (Figure 3A); this segregation is not apparent from single-nucleotide polymorphism (SNP) analysis of core gene sequences (Figure 1). Three strains of lineage III clustered closely by PCA, while HK_907, HK_973 and PN_696 were more individually positioned (Figure 3A). Again, this pattern of accessory gene content is not reflected in the SNP analysis of core gene sequences (Figure 1).

Lineage-specific gene homologs were abstracted from the Roary output, and the annotated genes (excluding hypotheticals) are listed in Appendix A. Thirty-seven gene homologs (14 annotated, 23 hypothetical) were detected in 23 lineage I strains and not in strains of other lineages; the corresponding numbers are 16 annotated genes only present in 11 lineage II strains, and 20 annotated genes only present in six lineage III strains. Several unexpected associations are observed, such as homologs of the multidrug exporter MdtA restricted to lineage II, and several CRISPR-associated nucleases confined to lineage II or III. Clearly, phenotypic and pathobiological significance must be addressed in biologic experiments, and the true relationship between marker genes and lineages awaits analysis of a larger number of strains. Nevertheless, the existence of lineage-specific marker genes encourages the development of lineage-specific PCRs that will be simpler and more informative than the currently employed serotype-specific PCRs. 

## 3. Discussion

*A. actinomycetemcomitans* was linked to aggressive periodontitis in 1976, and this association was supported by elevated serum antibodies in patients [5]. Three distinct bacterial surface antigens were identified by 1983, and this typing has remained the cornerstone of the initial characterisation of cultured strains. More advanced methods for dissection of the population structure [8,9,10] have not gained acceptance, although multilocus sequence typing (MLST)holds promise as a general, versatile typing method. WGSs have unequivocally shown that serotyping is inadequate for assessment of the phylogenetic positioning of a clinical strain.

We performed whole genome sequencing of *A. actinomycetemcomitans* cultured from blood stream infections. By serotype, the population resembled oral strains from our region with almost equal proportions of the three dominant serotypes. However, comparison of WGSs revealed some interesting observations. First, the serotype e strain PN_561 was not related to common serotype e strains within the species but belonged to an outgroup that has been designated clade e’ [13]. Aberrant strains of serotype e also deviate by 16S rRNA [9] and MLST [10,23], but distinctive phenotypic markers have not yet been described. Assessment of overall genome relatedness by ANI and in silico DDH positioned PN_561 and the reference clade e’ outgroup strain outside the species boundary. These strains are negative for the cytolethal distending toxin genes but encode the *tad* cluster that is decisive for autoaggregation and adherence to a wide range of solid surfaces. Strain HK_921 of the clade e’ outgroup was included in the investigation of *Aggragatibacter* strains that resulted in the description of the new species *Aggregatibacter kilianii*, and the difference between HK_921 and the other strains of *A. actinomycetemcomitans* was similar to or exceeded the difference between *Aggregatibacter kilianii* and *Aggregatibacter aphrophilus* [24]. Valid publication of bacterial names generally requires key phenotypic tests for discriminatory purposes, while identification by matrix-assisted laser desorption/ionisation time-of-flight (MALDI-TOF) mass spectrometry is only dependent on the composition of mass spectre in the database. The frequent detection of these outgroup strains in clinical samples may give impetus to taxonomic rearrangements in genus *Aggregatibacter*; the clinical significance of the clade e’ outgroup is emphasised by strain PN_561 being the cause of infective endocarditis.

Second, strain PN_696 of serotype a was neither related to other strains of this serotype nor to any previously deposited WGSs of *A. actinomycetemcomitans*. We were able to identify five additional strains cultured from the oral cavity with high genotypic resemblance to PN_696. In contrast to the clade e’ outgroup, aberrant serotype a strains were positioned inside the *A. actinomycetemcomitans* species boundary. Accepted and used designations are helpful, and clade a’ would be in line with a recent description of the population structure of *A. actinomycetemcomitans* by WGSs [13]. We suggest a different term. The population structure of *A. actinomycetemcomitans* is more accurately described by a division of the species into three phylogenetic lineages I–III. Although clearly separate entities, this designation would bundle serotype b and c strains into a common lineage. Strains of five separate serotypes are present in lineage II, while a subset of serotype a strains constitute lineage III. 

## 4. Materials and Methods

### 4.1. Bacterial Strains and DNA Accession Numbers

Twenty-nine strains of *A actinomycetemcomitans* were collected from blood stream infections from seven Danish departments of clinical microbiology. Initial analysis of WGSs revealed a peculiar sequence from a serotype a strain, and we were able to identify five additional oral strains with high resemblance to the invasive strain; these five strains were included in the study, as was the type strain NCTC 9710 of serotype c. Representative WGSs of serotypes other than c were downloaded from the public databases, including an additional serotype e strain allocated to the outgroup tentatively designated clade e’ [13]. Appendix A lists the origin, host characteristics, and accession numbers of all investigated strains and sequences.

### 4.2. Identification and Phenotyping

All clinical strains were subjected to renewed identification by matrix-assisted, laser desorption/ionization time-of-flight mass spectrometry (MALDI-TOF) as described [24]. Serotyping by PCR was performed as previously described [25]. Natural competence was investigated by transformation assays [22] using donor DNA from the D7S *hns* mutant that carries the kanamycin resistance gene cluster from pUC4K in the *H-NS* gene [26]. In brief, 20 µl of a dense bacterial suspension (OD_600nm_ = 0.3) was spread in a small area (diameter of ~10 mm) on a brain-heart infusion (BHI) agar plate. After incubation for 2 h, 10 µl of donor DNA (~1 µg) was added, gently mixed by using a loop, and further incubated for an additional 6 h. The bacteria were collected with a cotton swap, suspended in 400 µl of BHI broth, and plated on selective media containing kanamycin 50 (µg/mL); in parallel, diluted samples were plated on chocolate agar plates. Colonies were counted after two and seven days, and the transformation frequency was the ratio of the number of transformants to the number of cells plated.

### 4.3. DNA Sequencing, Genome Assembly, and Analysis

DNA libraries were prepared from 200 ng of genomic DNA with a Sciclone NGS robot (PerkinElmer), using the QIAseq FX DNA Library Kit (QIAGEN), according to the manufacturer’s protocol. Quality control of the libraries was conducted by on-chip electrophoresis (TapeStation, Agilent) and by Qubit (Thermofisher) concentration measurements. Dual-indexed paired-end sequencing (2 by 150 bp) was performed with an Illumina NextSeq 500 system (Illumina) aiming at 200 x coverage. Paired demultiplexed FASTQ files were generated using CASAVA software (Illumina), and initial quality control was performed using FastQC. Reads were assembled using Unicycler (version 0.4.7), an optimiser for SPAdes (version 3.13.9). Contigs with a length below 500 nt were disregarded. Draft assemblies of study strains plus FASTA files from reference strains downloaded from GenBank were annotated with Prokka [27]. Roary [28], a rapid, large-scale, prokaryote pan-genome analysis tool, was used with default settings for identification of core genes to create clusters of genes that share amino acid sequence similarity and coverage above a given threshold and orders strains by the presence or absence of orthologs. Core genes (present in all strains) were aligned with ClustalW and concatenated, before SNPs were called and evolutionary analyses conducted in MEGA X [29].

In silico DNA hybridisation between selected strains was performed with the Genome-to-Genome Distance Calculator (version 2.1), using standard settings and the recommended identity/high-scoring segment pair length calculation [17]. Average nucleotide identity (ANI) values of draft genomes were calculated using online tools (http://www.ezbiocloud.net/sw/oat) [19]; additionally, ANI values of concatenated *A. actinomycetemcomitans* core genes were calculated with Panito (version 0.0.2b1) (https://github.com/sanger-pathogens/panito). Principal component analysis (PCA) of binary absence/presence gene matrix from Prokka was computed in R with built-in packages. 

## Figures and Tables

**Figure 1 pathogens-08-00256-f001:**
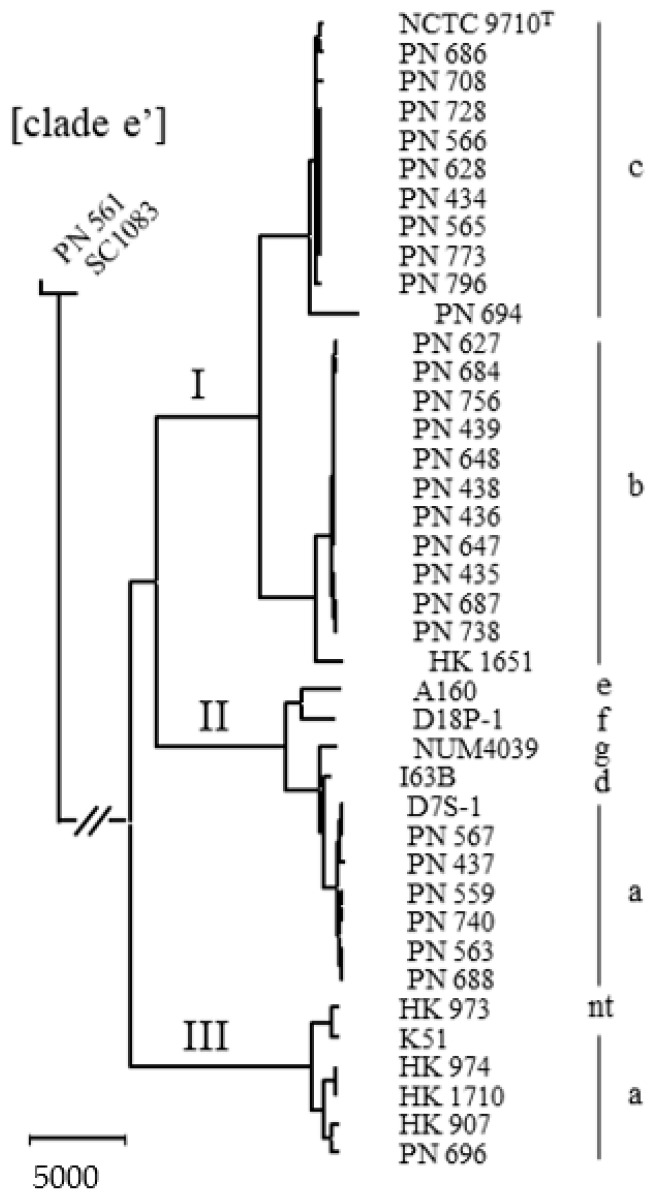
Neighbour-joining dendrogram of *Aggregatibacter actinomycetemcomitans* based on 1146 concatenated core genes (1,104,001 nucleotides) of 42 whole genome-sequenced strains. The type strain is designated with a superscript T. Twenty-nine strains were from cases of bacteraemia (designated PN), five oral strains were included to describe lineage III, and seven WGSs were downloaded from Genbank; see Appendix A for further description and origin of strains. The outgroup (strains PN_561 and SC1083) reduces the number of core genes and should probably be excluded from the species. Serotypes and phylogenetic lineages are shown; nt, non-typeable by immunodiffusion with antisera [8]. The bar represents 5000 residue substitutions.

**Figure 2 pathogens-08-00256-f002:**
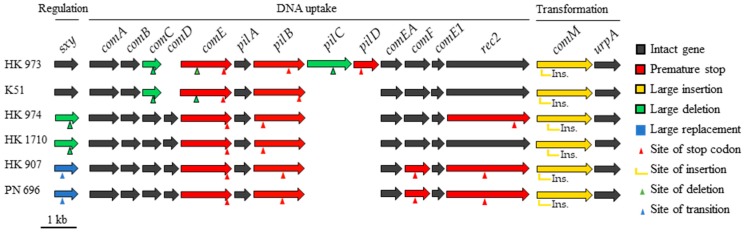
Intact and inactivated competence genes from lineage III strains using strain HK_1651 as reference. Red genes have premature stop codons caused by single base substitutions, single base deletions, or double nucleotide insertions. Yellow genes are disrupted by large insertions (11–28 Kb). Green genes have deletions (150–722 bp) of either the first (*sxy*, *pilC*, *comE*) or the last part (*comC*) of a gene. The first 73 amino acids of gene *sxy* in strains HK_907 and PN_696 are replaced with an unrelated coding sequence (blue genes). Green and blue arrows mark the transition from competence gene to unrelated sequence, or vice versa.

**Figure 3 pathogens-08-00256-f003:**
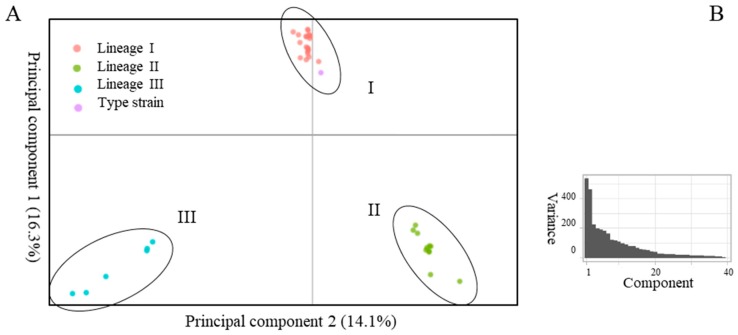
(**A**), principal component analysis of presence/absence of accessory homologs in 40 strains of *A. actinomycetemcomitans*. (**B**), line plot of the eigenvalues of factors or principal components in the analysis. The two principal components depicted in 3A comprise 30% of the sum of variances of all individual principal components (3B).

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
