# Peer review of "Whole Genome Sequencing of Aggregatibacter actinomycetemcomitans Cultured from Blood Stream Infections Reveals Three Major Phylogenetic Groups Including a Novel Lineage Expressing Serotype a Membrane O Polysaccharide"

_pathogens, 2019, doi:10.3390/pathogens8040256_

Round 1
Reviewer 1 Report
The authors investigated whole genome sequencing of A.actinomycetemcomitans and found that serotype did not match WGS.
There are comment to authors.
#1 Patient s’ status of periodontal disease are needed at least assessment of chronic or aggressive periodontitis.
#2 Line 174, WGS have unequivocally … a clinical strain. To detect phylogenetic positioning is necessary to know outline of the bacteria, however, for periodontal disease or A.a related systemic disease, it is inadequate why genotyping is needed than assessment of serotype. PN_561 was not related to common serotype. It is interesting, however, it does not have cytolethal toxin (line 184), hence for clinician it is unclear what is rationale to use WGS, what is advantage to use WGS. This has to be discussed clearly. Comment #1 may help this issue.
Author Response
Reviewer 1 (line numbers in [sharp brackets] refer to v2 of the manuscript)
The authors investigated whole genome sequencing of A.actinomycetemcomitans and found that serotype did not match WGS.
There are comment to authors.
#1 Patient s’ status of periodontal disease are needed at least assessment of chronic or aggressive periodontitis.
Reply: The reviewer pinpoints the periodontal status of bacteraemia patients. This is a relevant issue, because the focus of a bacteraemia case is not always identified, and larger series are not described for rarer pathogens such as HACEK organisms. At our institution, bacteraemia patients can be referred to a dentist/oral surgeon, who will remove teeth if periapical pathology or severe marginal periodontitis (pockets > 5 mm) are identified; however, this referral is only done for severe infections (infective endocarditis), and is of relatively recent origin. The present study covers a long period and multiple Hospitals, and this information cannot be retrieved; however, the reviewer’s comment is a helpful suggestion for future studies.
#2 Line 174, WGS have unequivocally … a clinical strain. To detect phylogenetic positioning is necessary to know outline of the bacteria, however, for periodontal disease or A.a related systemic disease, it is inadequate why genotyping is needed than assessment of serotype. PN_561 was not related to common serotype. It is interesting, however, it does not have cytolethal toxin (line 184), hence for clinician it is unclear what is rationale to use WGS, what is advantage to use WGS. This has to be discussed clearly. Comment #1 may help this issue.
Reply: We accept that WGS of pathogens is not essential for clinical care and outcome of infections, which are related to focus of infection and antimicrobial susceptibility. But when we embarked on the analysis, the interesting discovery was the unexpected population structure of A. actinomycetemcomitans. Consequently, the scientific focal point of the study was revised to accommodate this finding. The supplementary inclusion of non-bacteraemia strains was performed in order to display the genomic construction of the species, and the new lineage is described [Lines 66-7 and 77-82].
Reviewer 2 Report
In this paper the authors describe the classification of Aggregatibacter actinomycetemcomitans into three major phylogenetic groups. Traditional classification of A. actinomycetemcomitans into serotypes is insufficient for characterization. The authors have convincingly shown that the population structure of A. actinomycetemcomitans is more accurately described by a division of the species into three phylogenetic lineages I-III. They have also demonstrated the uneven distribution of accessory genes. The paper is well written and is an important advancement in the field.
Line 26: Change “lineages that differs” to “lineages that differ”
Line 38: “results in periodontitis and tooth loss in adolescents of African heritage”. This statement is misleading because it suggests that the disease happens only in people with African heritage.
Line 47: “prevalence proportion varied from 0% (Eikenella corrodens) to 100% (A. actinomycetemcomitans)”. Not clear what this means. If something is 100% then everything else must be 0 %.
Line 50: “Single studies have”. Not clear what this means.
Line 59: “strains between these groups”. Not clear what this means. Is this the right terminology to use?
Line 121: “invariable noncompetent”. I guess the authors mean, “invariably noncompetent”
Line 189: “Valid naming of new bacterial names”. I guess the authors mean, “Valid naming of new bacterial species”
Line 222: “dense bacterial solution”. I guess the authors mean “dense bacterial suspension”
Lines 116 – 129: I have two questions about the analysis of natural competence:
(1) The method described for natural transformation appear to be very simple and easy. Is transformation always so easy to demonstrate? Are the natural conditions for development of competence known? There have been reports in the past in which tfoX, the regulator of competence genes, had to be artificially cloned and expressed to induce competence. So is it possible that the authors have missed some strains that are actually transformable but not under the conditions in which they have been tested?
(2) If the authors already know the complete sequence of all these strains, why are they rationalizing transformation ability based on average genome size? Why not convincingly state if the competence genes are present or absent? Are these competence genes absent or interrupted in all serotypes that are non-competent?
Author Response
(line numbers in [sharp brackets] refer to v2 of the manuscript)
In this paper the authors describe the classification of Aggregatibacter actinomycetemcomitans into three major phylogenetic groups. Traditional classification of A. actinomycetemcomitans into serotypes is insufficient for characterization. The authors have convincingly shown that the population structure of A. actinomycetemcomitans is more accurately described by a division of the species into three phylogenetic lineages I-III. They have also demonstrated the uneven distribution of accessory genes. The paper is well written and is an important advancement in the field.
Line 26: Change “lineages that differs” to “lineages that differ”
Reply: Spelling has been corrected [L. 27]
Line 38: “results in periodontitis and tooth loss in adolescents of African heritage”. This statement is misleading because it suggests that the disease happens only in people with African heritage.
Reply: The JP2 clone probably emerged as a distinct genotype in Africa but has since spread to other populations. To avoid misunderstandings, the referral to “.. in adolescents of African heritage” has been removed [L. 38]
Line 47: “prevalence proportion varied from 0% (Eikenella corrodens) to 100% (A. actinomycetemcomitans)”. Not clear what this means. If something is 100% then everything else must be 0 %.
Reply: The sentence has been changed to: “ .. although the association with endocarditis ranged from 0 of 11 cases (Eikenella corrodens) to 18 of 18 cases (A. actinomycetemcomitans).” [L. 46-7]
Line 50: “Single studies have”. Not clear what this means.
Reply: “Single” has been changed to “Other” [L. 50]
Line 59: “strains between these groups”. Not clear what this means. Is this the right terminology to use?
Reply: Sentence has been revised: “.. while comparison of strains belonging to separate groups disclosed significant differences (ANI ..” [L. 58-9].
Line 121: “invariable noncompetent”. I guess the authors mean, “invariably noncompetent”
Reply: Spelling has been corrected [L. 118].
Line 189: “Valid naming of new bacterial names”. I guess the authors mean, “Valid naming of new bacterial species”
Reply: The sentence has been changed to comply with the International Code of Nomenclature of Prokaryotes (“Valid publication of bacterial names..”) [L. 198].
Line 222: “dense bacterial solution”. I guess the authors mean “dense bacterial suspension”
Reply: The sentence has been corrected [L. 230].
Lines 116 – 129: I have two questions about the analysis of natural competence:
(1) The method described for natural transformation appear to be very simple and easy. Is transformation always so easy to demonstrate? Are the natural conditions for development of competence known? There have been reports in the past in which tfoX, the regulator of competence genes, had to be artificially cloned and expressed to induce competence. So is it possible that the authors have missed some strains that are actually transformable but not under the conditions in which they have been tested?
(2) If the authors already know the complete sequence of all these strains, why are they rationalizing transformation ability based on average genome size? Why not convincingly state if the competence genes are present or absent? Are these competence genes absent or interrupted in all serotypes that are non-competent?
Reply: The experiments are extracted from an ongoing investigation involving a larger number of strains, and using both the kanamycin-cassette assay plus transfer of mutational resistance generated in two separate genes. To keep focus on the new Lineage (III), and because transformation efficiency was supported by genomic analysis, only the simple transformation assay was referred to, and only for the six strains of Lineage III. To further strengthen the argument, we have included a new Figure (new #2) depicting the known competence genes and regulators, including tfoX (sxY), described in [L. 127-9].
The argument of linkage between average genome size and competence for transformation was originally suggested by Jorth and Whiltely (2012). The priority is maintained [L. 120-22], but the challenge to the relationship between competence and genome size, at least for lineage III, is highlighted [L. 129-31]:
Reviewer 3 Report
Overall, this is a very well-written manuscript describing straightforward (but well-designed) research project about the sequencing and characterization of the whole genome of Aggregatibacter actinomycetemcomitans. A few minor changes are described below:
Line 26: “into three lineages that differs in ---” should be changed to “into three lineages that differ in---”
Lines 71-73: These sentences must be rephrased for clarity. “Serotyping by PCR identified seven serotype a, 11 serotype b, 10 serotype c, and one serotype e strain; this distribution is similar to the almost equal proportions of the three dominant serotypes observed among oral strains in Scandinavia [16].” In its current form, it’s not clear how the serotype distribution observed by the authors is correlated with the observations made in the review by Rylev and Killan (citation 16)
Line 165: “Nevertheless” is one word.
Lines 201-202: Not clear what the authors mean by “We suggest a term with a different connotation.”
Author Response
(line numbers in [sharp brackets] refer to v2 of the manuscript)
Overall, this is a very well-written manuscript describing straightforward (but well-designed) research project about the sequencing and characterization of the whole genome of Aggregatibacter actinomycetemcomitans. A few minor changes are described below:
Line 26: “into three lineages that differs in ---” should be changed to “into three lineages that differ in---”
Reply: Spelling has been corrected [L. 27].
Lines 71-73: These sentences must be rephrased for clarity. “Serotyping by PCR identified seven serotype a, 11 serotype b, 10 serotype c, and one serotype e strain; this distribution is similar to the almost equal proportions of the three dominant serotypes observed among oral strains in Scandinavia [16].” In its current form, it’s not clear how the serotype distribution observed by the authors is correlated with the observations made in the review by Rylev and Killan (citation 16)
Reply: The sentence has been simplified: “.. is similar to the observed prevalence among oral strains in Scandinavia [16 and references therein]” [L. 72-4]
Line 165: “Nevertheless” is one word.
Reply: Spelling has been corrected [L. 174].
Lines 201-202: Not clear what the authors mean by “We suggest a term with a different connotation.”
Reply: The sentence has been changed: “We suggest a different term” [L. 209].
Round 2
Reviewer 1 Report
The authors responsed accurately to the review comments.
Please state about informed consent and ethic approvement to use patients' sbject matter including clinical strain used for the present study.
Author Response
Statements regarding ethics and funding have been appended to the paper